# Characterization of Mitochondrial Bioenergetics in Preeclampsia

**DOI:** 10.3390/jcm10215063

**Published:** 2021-10-29

**Authors:** Ramana Vaka, Evangeline Deer, Mark Cunningham, Kristen M. McMaster, Kedra Wallace, Denise C. Cornelius, Lorena M. Amaral, Babbette LaMarca

**Affiliations:** 1Department of Pharmacology, Physiology & Toxicology, Center for Excellence in Cardiovascular and Renal Research, University of Mississippi Medical Center, Jackson, MS 39216, USA; venkataramanavaka71@gmail.com (R.V.); edeer@umc.edu (E.D.); mwcunningham@umc.edu (M.C.); lmamaral@umc.edu (L.M.A.); 2Department of Obstetrics and Gynecology, Center for Excellence in Cardiovascular and Renal Research, University of Mississippi Medical Center, Jackson, MS 39216, USA; kristen.m.mcmaster@gmail.com (K.M.M.); kwallace2@umc.edu (K.W.); 3Department of Emergency Medicine, University of Mississippi Medical Center, Jackson, MS 39216, USA; dcornelius@umc.edu

**Keywords:** mitochondria, electron transport chain, oxidative stress, preeclampsia, placenta, reactive oxygen species

## Abstract

Preeclampsia (PE) is characterized by new onset hypertension during pregnancy and is associated with oxidative stress, placental ischemia, and autoantibodies to the angiotensin II type I receptor (AT1-AA). Mitochondrial (mt) dysfunction in PE and various sources of oxidative stress, such as monocytes, neutrophils, and CD4 + T cells, have been identified as important players in the pathophysiology of PE. We have established the significance of AT1-AA, TNF-α, and CD4 + T cells in causing mitochondrial (mt) dysfunction in renal and placental tissues in pregnant rats. Although the role of mt dysfunction from freshly isolated intact placental mitochondria has been compared in human PE and normally pregnant (NP) controls, variations among preterm PE or term PE have not been compared and mechanisms contributing to mt ROS during PE are unclear. Therefore, we hypothesized PE placentas would exhibit impaired placental mt function, which would be worse in preterm PE patients than in those of later gestational ages. Immediately after delivery, PE and NP patient’s placentas were collected, mt were isolated and mt respiration and ROS were measured. PE patients at either < or >34 weeks gestational age (GA) exhibited elevated blood pressure and decreased placental mt respiration rates (state 3 and maximal). Patients delivering at >34 weeks exhibited decreased Complex IV activity and expression. Placental mtROS was significantly reduced in both PE groups, compared to NP placental mitochondria. Collectively, the study demonstrates that PE mt dysfunction occurs in the placenta, with mtROS being lower than that seen in NP controls. These data indicate why antioxidants, as a potential target or new therapeutic agent, may not be ideal in treating the oxidative stress associated with PE.

## 1. Introduction

Preeclampsia (PE) is defined by the development of new-onset hypertension with organ dysfunction with or without proteinuria after 20 weeks of gestation and affects approximately 10% of pregnancies worldwide [1,2,3,4]. Unfortunately, there are no effective treatments available for PE, except for the delivery of the fetus and placenta. Research has suggested that the addition of antioxidants during pregnancy is essential to the improvement of antioxidant capacity of the placenta and fetus and beneficial for overall maternal health. Yet, there is still more to reveal in the molecular mechanisms underlying the pathogenesis of PE to establish targeted therapeutic options. Oxidative stress is an imbalanced state between reactive oxygen species (ROS) and antioxidant reserves within the cell and has been shown to be associated with the pathology of PE [5,6,7,8]. The term ROS is used to describe highly reactive free radicals, such as hydrogen peroxide (H_2_O_2_), superoxide (O^−2^), hydroxyl radical (·OH), and peroxynitrite (ONOO^−^), which cause substantial damage to DNA, RNA, and proteins, and result in cellular dysfunction and death [9]. Mitochondria are dynamic cellular organelles that are popularly known as the power houses of the cell. ETC, located on the inner mitochondrial membrane, is composed of four complexes (Complex I-IV), and is coupled to Complex V (ATP synthase) to generate energy (ATP) via oxidative phosphorylation (OXPHOS). During OXPHOS, the oxidation of reducing equivalents (NADH and FADH2) at complex I (NADH dehydrogenase) and II (succinate dehydrogenase) release a pair of electrons, which, in turn travel through ETC complexes by a series of redox reactions carried out by electron carriers located within the ETC. This electron transfer causes proton translocation from matrix into the inner membrane space leading to the development of proton motive force (PMF) which in turn drives Complex V to phosphorylate ADP to form ATP. While Complex I and II are the electron entry ports, complex IV is where the oxygen consumption occurs. Importantly, Complex I is a major known site of ROS generation within mitochondria. Although, minimal ROS production is shown to have a physiological significance, excessive amounts of ROS generation (occurs often due to mitochondrial dysfunction) leads to cell death [10,11]. 

Several studies have established the causative role of oxidative stress in PE and associate it with reduced ETC activity, placental mt morphological changes, and modifications in mt biogenesis and mtDNA in trophoblasts [12,13,14,15,16,17,18,19]. Although these and other studies validate the important role for impaired placental mt function in causing oxidative stress in PE, an effect of the impaired bioenergetics (mt respiration and mtROS) of placental mitochondria from preeclamptic women is not conclusive. We have previously shown the relevance of impaired mt function (respiration or oxidative stress) in causing hypertension in response to placental ischemia in a RUPP (reduced uterine perfusion pressure) rat model of PE [20] and have established a link of impaired placental mt dysfunction to endothelial mt dysfunction caused by circulating AT1-AA of women with PE [21], thus indicating the important role that impaired mt function plays during preeclampsia. Therefore, our objective was to determine if freshly isolated placental mt from PE patients would exhibit impaired mt function based on differences in their gestational age (PE < 34 weeks compared to PE >34 samples), compared to NP controls.

## 2. Materials and Methods

### 2.1. Materials and Reagents

Glutamate (49621)/ malate (M1000), succinate (14160), oligomycin (O4876), FCCP (C2920), rotenone (R8875), antimycin A (A8674), sucrose (84097), HEPES (H3375), EGTA (E3889), MOPS (M1254), KPi (795488), TMPD (T3134), NADH (10107735001), KCN (60178), decyl ubiquinone (D7911), DCPIP (D1878), and DNTB (D218200) were purchased from Sigma Aldrich (St. Louis, MO, USA). Amplex Red UltraRed Reagent (A36006) was purchased from Thermo Fisher Scientific (Waltham, MA, USA). Total OXPHOS cocktail antibody (MS604-300), VDAC (ab15895), and GAPDH antibody (ab8245) were purchased from abcam (Waltham, MA, USA). 

### 2.2. Patient Recruitment and Sample Collection

The participants were recruited from the Department of Obstetrics and Gynecology, University of Mississippi Medical Center, Jackson, MS, USA. The University of Mississippi Medical Center Institutional Review Board approved the research protocol and all participants provided informed consent. 

All women included in the study were pregnant with singleton gestations undergoing either a vaginal or cesarean delivery performed for usual obstetric indications. Preeclampsia (PE) was defined as the development of de novo hypertension (≥140/90 mmHg) and proteinuria (>300 mg/24 h or +1 on repeat dipstick). Women with multiple gestations, fetal anomalies, or those diagnosed with gestational diabetes were excluded. Women with preexisting medical conditions such as chronic hypertension, diabetes, sickle cell disease, lupus, other inflammatory disorders, or sexually transmitted infections, and those that used tobacco or participated in other substance abuse during pregnancy were also excluded from the study.

The obstetric patients were admitted to the Winfred Wiser Hospital for Women and Infants for delivery due to regular obstetric indications. Patients included in the study consented to having their blood drawn prior to delivery, and for the use of their discarded placental tissue immediately following delivery. The patients’ blood was collected via venipuncture in collection tubes (BD Vacutainer, Fisher) for serum. Per the manufacturer instructions, the samples remained at room temperature for two hours to clot and then centrifuged for 10 min at 4 °C at 3200 rpm with serum subsequently stored at −80 °C [20,21]. Fresh mitochondria were immediately isolated from placentas after delivery. Multiple 2–3-mm^3^ placental tissue explants were collected from the maternal side of the placenta and stored at −80 °C for future use in Western blots for complex expression and for complex activity assays. Study arms included Normal Pregnant Control placentas (NP Controls, *n* = 30) and preeclamptic patient placentas that were either greater than or less than 34 weeks gestation (PE < 34 weeks GA (*n* = 10) PE > 34 weeks GA (*n* = 13)).

### 2.3. Isolation of Intact Mitochondria from Human Placenta

Intact mitochondria were isolated from each human placenta using differential centrifugation methods. Succinctly, three placental cotyledons were isolated from the maternal side of the placenta. The placental pieces were immediately placed in ice cold MSHE buffer (210 mM mannitol, 70 mM sucrose, 1 mM EDTA, 5 mM HEPES, pH 7.4) for rinsing. The washed placental tissue was minced, and the minced tissue was washed and filtered over a thin surgical gauze (repeated three times). The rinsed tissue was transferred to a Potter–Elvehjem homogenizer and homogenized. The homogenate was centrifuged for 10 min at 4 °C at 1500× *g*. The supernatant was collected and centrifuged at 12,000× *g* for 10 min at 4 °C. The pellet was collected, resuspended in 2 mL of MSHE buffer and centrifuged at 12,000× *g* for 10 min at 4 °C. The final pellet was isolated and re-suspended in 400 µL of MSHE buffer and was immediately used for bioactivity assays. Mitochondrial protein was quantitated by Bicinchonic acid assay (BCA). A portion of the mitochondrial sample was flash-frozen and stored at −80 °C for ETC complex enzyme activity and expression assays.

### 2.4. Measurement of ETC Activity

Complex IV (cytochrome c oxidase) activity was measured using Oxygraph 2K. The reaction mixture, containing 2 mL of Tris buffer (50 mM, pH 7.4) along with ascorbate (3 mM), *N*,*N*,*N*′,*N*′-tetramethyl-*p*-phenylenediamine (TMPD) (0.3 mM), horse heart cytochrome c (40 μM), was added to the Oxygraph chamber. The reaction was started by adding the mitochondrial sample (30 µL). Oxygen consumption before adding the mitochondria was subtracted to correct for nonmitochondrial oxygen consumption. Complex IV activity was normalized to the amount of mitochondrial protein and expressed as nmol of electrons transferred/min/mg protein. 

### 2.5. Measurement of ETC Expression by Immunoblotting

The expression of ETC complexes in the isolated mitochondria was examined by Western blotting. Briefly, mitochondrial samples were electrophoresed at 200 V for 45 min on criterion gels and transferred to nitrocellulose membrane using a Trans-Blot Turbo Transfer System (Biorad, Irvine, CA, USA). After transfer, the membrane was blocked for 1 h using odyssey blocking buffer. The membrane was then incubated with the primary antibodies, total OXPHOS antibody cocktail (6.0 µg/mL), and VDAC (Voltage Dependent Anion Channel), overnight, at 4 °C. The following day, the membrane was washed with TBST three times and incubated with the appropriate secondary antibody for 45 min at room temperature. The membrane was then washed twice with TBST and with TBS once before scanning using a LICOR system. The protein band densities were analyzed using LICOR analysis software.

### 2.6. Measurement of Respiration in Isolated Mitochondria

Oroboros Oxygraph 2K was used to perform respiration measurements in the isolated placental mitochondria. Respiration measurements were performed with 30 µL of isolated mitochondrial sample in 2 mL of respiration buffer (100 mM KCl, 5 mM KPi, 1 mM EGTA, BSA, 1 mg/mL, 50 mM MOPS, pH 7.4). Substrates, glutamate (10 mM)/malate (2 mM), for complex I mediated respiration, or succinate (10 mM), for complex II mediated respiration, were added to record the state 2 respiration. State 3 (coupled respiratory state) and maximal respiration (uncoupled respiration) rates were measured by the addition of ADP (5 mM) and FCCP (0.5 µM), respectively, to the Oxygraph-2K chamber. Finally, rotenone (0.5 µM) and antimycin A (2.5 µM) were injected to record non-mitochondrial respiration. This rate was subtracted from state 3 or maximal rate to correct for non-mitochondrial respiration. The collected data were normalized to the amount of mitochondrial protein and expressed as pmol of oxygen consumed/s/mg protein

### 2.7. Measurement of Mitochondrial ROS Production in Isolated Mitochondria

Mitochondrial ROS (H_2_O_2_) production in the isolated mitochondria was measured by Amplex red assay. H_2_O_2_ oxidizes the non-fluorescent amplex red-to-red fluorescent oxidation product, resorufin, in the presence of horseradish peroxidase (HRP). Briefly, mitochondria (0.4 mg/mL) were incubated in 96-well plate with respiration buffer (100 mM KCl, 5 mM Kpi, 1 mM EGTA, 1 mg/mL BSA fatty acid free, and 50 mM MOPS, pH 7.4), superoxide dismutase (SOD, 40 U/mL), HRP (4 U/mL), succinate (10 mM), and a complex II substrate. The reaction was started by adding Amplex red (10 µM) to the wells. Real-time change in fluorescence, indicative of H_2_O_2_ production, was recorded using a plate reader at the 555/581 nm excitation/emission wavelengths for 30 min at 25 °C. Appropriate blanks, without Amplex red or mitochondrial protein, were included in the assay.

### 2.8. Statistical Analysis

All data are expressed as mean ± standard error of mean (SEM). A one-way ANOVA was performed for statistical comparisons between NP, PE < 34 weeks, and PE > 34 groups. A student’s t test was performed to determine differences between NP and either PE < 34 weeks or PE > 34 weeks groups. A value of *p* < 0.05 was considered statistically significant.

## 3. Results

### 3.1. Patient Demographics

The study included 30 normal pregnant (NP) and 23 women with preeclampsia (*n* = 10) PE < 34 weeks and (*n* = 13) PE > 34 weeks. Demographic and patient delivery information can be found in Table A1. MAP in PE patients was significantly higher at delivery (PE < 34 weeks (107.6 ± 3.34 mmHg, *p* < 0.05, *n* = 10) and PE > 34 weeks (96.33 ± 3.74 mmHg; *p* < 0.05, *n* = 13)) in comparison to NP controls (86.4 ± 1.93 mmHg, *n* = 30) (Figure 1). Furthermore, there was also a significant increase in MAP at delivery of PE > 34 weeks compared to PE < 34 weeks (*p* < 0.05). 

Gestational age was lower in PE < 34 weeks (30.30 ± 0.81 weeks, *p* < 0.05, *n* = 10) and PE > 34 weeks (37.83 ± 0.44 weeks, *n* = 13; (*p* < 0.05 vs. PE < 34 weeks)) compared with NP women (37.51 ± 0.19 weeks, *n* = 30). Fetal weight was also lower in PE < 34 weeks (1516 ± 108.1 g, *p* < 0.05, *n* = 10) and PE > 34 weeks (3120 ± 153.3 g, *n* = 13) compared with NP women (3169 ± 96.9 g, *n* = 30). There was also a significant increase in the fetal weight of PE > 34 weeks compared to PE < 34 weeks (*p* < 0.05).

### 3.2. Placental Mitochondrial Electron Transport Chain Complex IV Activity and Expression Are Reduced in Preeclampsia

Placental Complex IV activity was significantly reduced in PE > 34 weeks (*n* = 9) vs. NP (*n* = 10) mitochondria (Figure 2A, 141.4 ± 18.2 vs. 238.5 ± 24.13 nmol e^−^/min/mg, *p* < 0.05). Furthermore, Western blot analysis demonstrated that Complex IV expression was also significantly reduced in PE mitochondria compared with normotensive controls (Figure 2B,C, 0.505 ± 0.09 vs. 1.344 ± 0.080, Complex IV/VDAC, *p* < 0.05). Our attempts at collecting Complex IV activity with samples from PE < 34 weeks were unsuccessful. The expression of other complexes was not different between PE, at either gestational age, and NP patients. 

### 3.3. Mitochondrial Respiration Is Reduced in Preeclamptic Placental Mitochondria

Preeclampsia was associated with a trend toward decreased respiration in all respiration states measured in both Complex I- and Complex II-mediated respiration (Figure 3 and Figure 4). In placental mitochondrial Complex I, PE patients’ state 3 [PE < 34 weeks (134 ± 42 pmol O_2_/s/mg, *n* = 4) and PE > 34 weeks (98 ± 24 pmol O_2_/s/mg, *n* = 5, *p* < 0.05)] (Figure 3A) and maximal respiration rates (PE < 34 weeks (274 ± 106 pmol O_2_/s/mg, *n* = 4) and PE > 34 weeks (168 ± 51 pmol O_2_/s/mg, *n* = 5) (Figure 3B)) were reduced compared to NP women (293 ± 56 pmol O_2_/s/mg, *n* = 7; 457 ± 131 pmol O_2_/s/mg, *n* = 7). In placental mitochondrial Complex II, PE patients’ state 3 (PE < 34 weeks (222 ± 90 pmol O_2_/s/mg, *n* = 4, *p* < 0.05) and PE > 34 weeks (178 ± 14 pmol O_2_/s/mg, *n* = 5, *p* < 0.05) (Figure 4A)) and maximal respiration rates (PE < 34 weeks (194 ± 54 pmol O_2_/s/mg, *n* = 4, *p* < 0.05) and PE > 34 weeks (272 ± 63 pmol O_2_/s/mg, *n* = 5) (Figure 4B)) were reduced compared to NP women (441 ± 38 pmol O_2_/s/mg, *n* = 7; 404 ± 43 pmol O_2_/s/mg, *n* = 7).

### 3.4. Mitochondrial ROS Is Reduced in Preeclamptic Placental Mitochondria

In PE mitochondria, mtROS was significantly reduced at either gestational age of PE (PE < 34 weeks (29 ± 4, FLU, *n* = 4, *p* < 0.05; PE > 34 weeks (72 ± 3, FLU, *p* < 0.05, *n* = 5)) in comparison to NP mitochondria (100 ± 6.1, FLU, *n* = 5) (Figure 5). 

## 4. Discussion

Preeclampsia is a frequent complication of pregnancy, but unfortunately its cause is unknown. Oxidative stress and an inflammatory state are well-known features of PE. Although, the role of oxidative stress in PE pathology has been extensively studied, this is the first study to demonstrate reduced bioenergetics (respiration and mtROS) from freshly isolated placental mitochondria from PE patients stratified by gestational age and compared to NP controls. Our findings reveal that mitochondrial respiration is reduced, along with reductions in both Complex IV function and expression in PE placentas, compared with NP placentas. In conjunction with data published by McCarthy et al. [22], our recent publication from Deer et al. [21], demonstrated that soluble factors released in association with placental dysfunction cause vascular mtROS and decrease respiration in human umbilical vein endothelial cells (HUVECs) exposed to PE sera stratified by gestational age, compared to NP sera. Collectively, these data support the hypothesis that mt bioenergetics is impaired among PE patients which may be worse at deliveries during earlier gestational ages. 

Oxidative stress is the result of an imbalance between the production of reactive oxygen species (ROS) and antioxidant defenses in the cell [23]. While oxidative stress is shown to be associated with normal pregnancy physiology [5,7], the degree of oxidative stress is exacerbated further in pregnancy complications such as PE, gestational diabetes, and anemia [5,7]. As the placenta is a vital organ not only for fetal development but also because of its influence on maternal physiology, we sought to investigate the pathophysiological changes in preeclamptic placental bioenergetics. Mitochondria are the known major source of ROS generation within the cell. During mitochondrial dysfunction, mitochondria generate excessive levels of ROS often in association with reduced oxygen consumption [11]. In addition to mitochondria, NADPH oxidase and xanthine oxidase are the two other systems that were shown to mediate ROS generation in PE placenta [24,25]. NADPH oxidase’s role in causing oxidative stress in PE patients or animal models has been previously documented by multiple studies [24,25,26,27,28]. Specifically, we have shown that a PE rat model (RUPP) exhibited NADPH oxidase-mediated elevated placental ROS production [29], as well as impaired mitochondrial function [20,21]. Dikalov et al., demonstrated a cross talk between NADPH oxidase and mitochondria in mediating hypertension in an angiotensin-II mouse model of hypertension [30]. However, NADPH oxidases (or other ROS generating systems), in relation to mitochondria, are out of scope of this study, and, based on our strong preclinical work supporting mitochondria-mediated oxidative stress in preeclampsia, we sought to assess mitochondrial function and ROS generation in preeclamptic patients. The existing literature shows that mitochondrial dysfunction is evident in the preeclamptic placenta [12,13,14,16,17,18,21,31,32,33,34,35,36,37]. However, these published studies were largely focused on examining oxidative stress/antioxidant markers, and electron transport chain (ETC) complex activity in mitochondrial membranes or whole cells (i.e., trophoblasts) isolated from PE placentas. Thus, in this study we aimed to assess mitochondrial oxygen consumption and real-time ROS generation using intact coupled mitochondria isolated from placentas of preeclamptic (early and late gestational ages) patients and normal pregnant women. 

Firstly, we showed that frozen placental mitochondrial membranes from PE patients exhibit significant reductions in both the activity and expression of Complex IV (Figure 2A,B). However, we have not found any significant changes in the expression of other complexes of ETC (Figure 2C). Interestingly, Muralimanoharan et al. reported a reduction in Complex III or Complex I + III activities, and Complex I or Complex IV expressions in PE patients, with no differences found with respect to other complexes [31]. Further, Holland et al. showed that Complex II and III expressions were reduced with no change in Complex I, IV, or V expression in PE placentas in comparison to normal-pregnancy placentas [38]. This inconsistency in the ETC complex data is possibly due to variations in the preparation and handling of mitochondria across different research groups. This may have also been a reason why we were unsuccessful at collecting Complex IV data from the <34 weeks gestation PE cohort. Overall, Complex IV dysfunction has been consistently reported as being reduced in multiple independent studies, including ours [15,19,37]. Furthermore, our current study results suggest that the reduced Complex IV activity could be due to its low expression (Figure 2C)

Next, with the goal of understanding the bioenergetic profile of the preeclamptic placenta, we have isolated intact mitochondria using differential centrifugation. Given that isolated mitochondria contain low levels of reducing equivalents (NADH or FADH_2_), we have employed the addition of exogenous glutamate/malate (NADH) and succinate (FADH_2_) to assess respiration rates. Under glutamate/malate (Complex I-mediated) or succinate (Complex II-mediated), we measured state 3 (ADP induced) and maximal (FCCP induced) respiration rates. We found that placental mitochondria show a significantly lower rate of state 3 respiration (Complex I- or II-mediated) in both gestational ages of the assessed preeclamptic patients (Figure 3A and Figure 4A), suggesting a significant reduction in ATP production in PE mitochondria. Further, both Complex I- or II-mediated maximal respiration rates showed a trend towards reduction in late-gestational-age PE patients, while early-gestational-age PE patients showed a significant reduction in respiration rate (Complex II-mediated) (Figure 3 B and Figure 4B). These lowered maximal respiration rates indicate a compromised maximal electron transfer capacity of the ETC. Taken together, the reduced respiration rates can be explained by reduced Complex IV activity (Figure 2A) in PE mitochondria. Commensurate with our findings, miR-210, a micro-RNA implicated in preeclamptic pathology, has been shown to reduce respiration in trophoblasts upon transfection [31]. Contrarily, Mando et al. reported no differences in the respiration rates of the placental trophoblasts of PE patients in comparison to normally pregnant patients [39]. While this is an interesting finding, the authors utilized isolated cytotrophoblasts for their respiration measurements, hence, these findings do not represent a bioenergetic profile of a whole placenta. Since our mitochondrial preparations were made from a whole placental tissue, our results reflect the overall placental bioenergetic profile and explain why our findings differ from the Mando et al. study. 

Further, we have examined real-time mtROS production in isolated mitochondria. Generally, increased mtROS production leads to lipid peroxidation, damage to ETC and mtDNA, and ultimately lead to the initiation of apoptotic pathways [11]. In the current study, real-time measurement of mtROS (H_2_O_2_) in isolated mitochondria respiring under succinate was performed using Amplex Red assay. Our findings show a significant reduction in mtROS generation PE patients of both gestational groups (PE < 34 weeks and PE > 34 weeks) in comparison to NP (Figure 5). Generally, low respiration rates are often associated with increased mtROS production, as the unused oxygen gets utilized for superoxide production. However, H_2_O_2_ content will also depend on the systems that produce H_2_O_2_ from superoxide (MnSOD) or the systems that transform H_2_O_2_ into H_2_O (GPRx) or its conversion to other forms of ROS, such as hydroxyl radical (HO.) [11]. In agreement with our findings, Holland et al. showed that preterm PE placental mitochondria exhibit low H_2_O_2_ production in comparison to control pregnancies, and further showed that SOD expression in PE placental mitochondria was significantly low [38]. Hence, it could be that the low H_2_O_2_ levels seen in human preeclamptic placentas are a result of the low expression of systems that carry out the dismutation of superoxide. The discrepancy in H_2_O_2_ (mtROS) findings between humans and rodents is possibly due to the differential expression of enzymes that participate in mtROS metabolism. The superoxide, once produced from ETC, will undergo transformation into various forms of ROS, such as peroxynitrate, H_2_O_2_, and hydroxyl radical in the presence of matrix enzymes such as SOD, GPRx, and catalase. Hence, the changes in any of these enzymes will cause differential production of these ROS forms within the mitochondria. Thus, mt oxidative stress (due to high superoxide, peroxynitrate, or hydroxyl radical) can still manifest, despite low H_2_O_2_ levels. Hence, further studies utilizing simultaneous examination of various ROS forms in preeclamptic placental mt could unravel the intricate dynamics of ROS metabolism and oxidative stress in PE. 

## 5. Conclusions

Our study demonstrated that placenta from preeclamptic patients of both early and late gestational ages show compromised mitochondrial function with low levels of mtROS. While these are important findings in exploring mitochondrial bioenergetics in preeclampsia, the reduction in mtROS was a surprising finding to us. Based on our preclinical findings evaluating the therapeutic effect of mitochondrial targeted antioxidants and the extensive existing literature on the role of mitochondrial oxidative stress in preeclampsia, future studies exploring mtROS’s metabolic regulation will further advance our understanding of mitochondria-mediated damage in preeclampsia. 

## Figures and Tables

**Figure 1 jcm-10-05063-f001:**
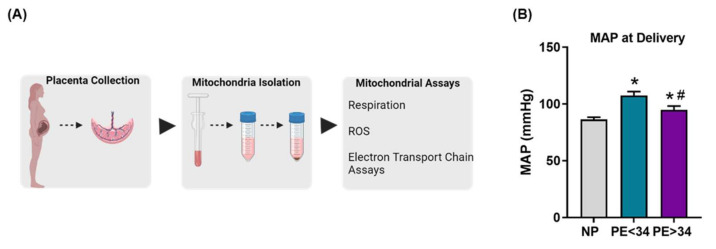
Conceptual figure showing the methodology utilized in the study (**A**), mean arterial pressure (MAP) at the delivery in normal pregnant (NP; gray bar) and preeclamptic patients of early (PE < 34; teal bar) and late gestational ages (PE > 34; purple bar) (**B**). MAP was elevated in both early and late gestational preeclamptic patients (PE < 34 weeks; *n* = 10 & PE > 34 weeks; *n* = 13) vs. normal pregnant (NP, *n* = 30). Further, there was a significant increase in MAP in early gestational patients (PE < 34 weeks) vs. late gestational patients (PE > 34 weeks) (**B**). The results represent means ± SEM. * *p* < 0.05 vs. NP, # *p* < 0.05 vs. PE < 34. ROS, reactive oxygen species; SEM, standard error of mean.

**Figure 2 jcm-10-05063-f002:**
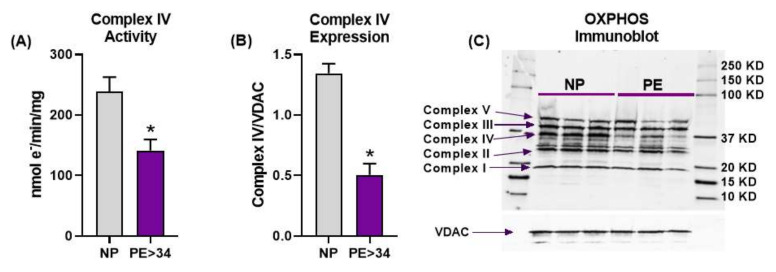
Placental mitochondrial electron transport chain (ETC) activity and expression. Oxygraph-2K cytochrome C oxidase assay showed Complex IV activity was significantly reduced in placental mitochondria of late gestational age preeclamptic patients (PE > 34 weeks, *n* = 9; purple bar) vs. normal pregnant (NP, *n* = 10; gray bar) (**A**). Western blot analysis showed Complex IV expression (*n* = 3) was significantly down regulated in placental mitochondria of late gestational age preeclamptic patients (PE > 34 weeks) vs. normal pregnant (NP, *n* = 3) (**B**). Immunoblotting performed using total OXPHOS cocktail antibody and voltage-dependent anion channel (VDAC, loading control) showing the protein bands for the ETC complexes (**C**). There was no difference observed in other complex expressions other than Complex IV between PE > 34 weeks vs. NP. The results represent means ± SEM. * *p* < 0.05 vs. NP. e^−^, electron; KD, kilodalton; mg, milligram; min, minute; nmol, nanomole; SEM, standard error of mean.

**Figure 3 jcm-10-05063-f003:**
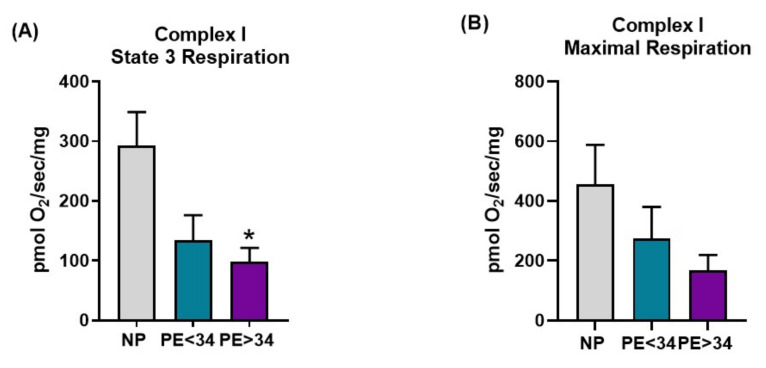
Complex I-mediated respiration in isolated mitochondria. Oxygraph-2K oxygen consumption assay showed Complex I placental mitochondrial state 3 respiration was significantly decreased in preeclamptic patients of late gestational age (PE > 34 weeks, *n* = 5; purple bar) and a trend towards reduction in preeclamptic patients of early gestational age (PE < 34 weeks, *n* = 4; teal bar) vs. normal pregnants (NP, *n* = 7; gray bar) (**A**). Further, placental mitochondrial maximal respiration showed a trend towards reduction in preeclamptic patients of both gestational ages (PE < 34 weeks, *n* = 4 and PE > 34 weeks, *n* = 5) vs. normal pregnants (NP, *n* = 7) (**B**). The results represent means ± SEM. * *p* < 0.05 vs. NP. pmol, picomoles; O_2_, oxygen, mg, milli grams, sec, second; SEM, standard error of mean.

**Figure 4 jcm-10-05063-f004:**
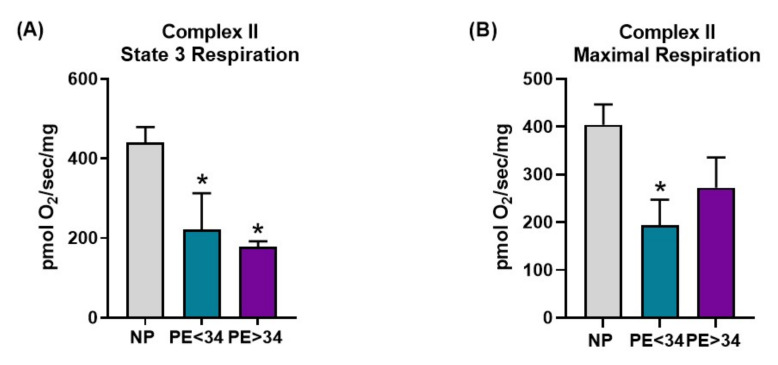
Complex II-mediated respiration in isolated mitochondria. Oxygraph-2K oxygen consumption assay showed Complex II mediated placental mitochondrial state 3 respiration was significantly reduced in both preeclamptic patients of both gestational ages (PE < 34 weeks, *n* = 4; teal bar and PE > 34 weeks, *n* = 5; purple bar) vs. normal pregnants (NP, *n* = 7; gray bar) (**A**). Further, placental mitochondrial maximal respiration was reduced in preeclamptic patients of early gestational age (PE < 34 weeks, *n* = 4) and showed a trend in towards reduction in preeclamptic patients of late gestational age (PE > 34 weeks, *n* = 5) vs. normal pregnants (NP, *n* = 7) (**B**). The results represent means ± SEM. * *p* < 0.05 vs. NP. pmol, picomoles; O_2_, oxygen, mg, milli grams, sec, second; SEM, standard error of mean.

**Figure 5 jcm-10-05063-f005:**
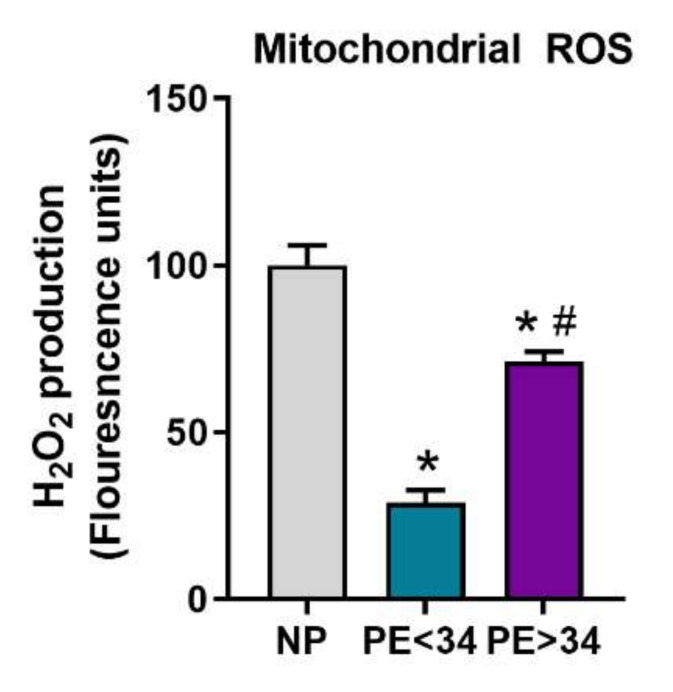
Mitochondrial ROS in isolated placental mitochondria. Amplex Red assay showed mitochondria from preeclamptic patients of both gestational ages (PE < 34 weeks, *n* = 4; teal bar and PE > 34 weeks, *n* = 5; purple bar) exhibit reduced H_2_O_2_ production vs. normal pregnant (NP, *n* = 5; gray bar). Further, H_2_O_2_ was found to be significantly lower in preeclamptic patients of early gestational age (PE < 34 weeks, *n* = 4) vs. preeclamptic patients of late gestational age (PE > 34 weeks, *n* = 5). The results represent means ± SEM. * *p* < 0.05 vs. NP, # *p* < 0.05 vs. PE < 34. H_2_O_2,_ hydrogen peroxide; ROS, reactive oxygen species, SEM, standard error of mean.

## Data Availability

The data presented in this study are available on request from the corresponding author.

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
