# Peer review of "Characterization of Mitochondrial Bioenergetics in Preeclampsia"

_jcm, 2021, doi:10.3390/jcm10215063_

Round 1
Reviewer 1 Report
In this study conducted by Vaka et al, the authors investigated the impaired mitochondrial dysfunction in preeclamptic placentas. The study addresses an important issue and will be of interest for scientists in this field of research. However, there are some points which need to be addressed or clarified.
- The introduction needs to be improved. As the study investigates the functionality of different complexes in ETC, a brief inclusion of functioning of ETC and unique roles of different complexes will be important for the readers to better understand the results and discussion.
- In the original protocol of isolation of trophoblast mitochondria, described by Fisher et al, they spun the supernatant at 4000 g for cytotrophoblast and 12000 g for syncytiotrophoblast isolation, respectively. However, the authors in this study used only 12000 g indicating that the pellet to isolate mt had both cyto and syncytiotrophoblast. What was the rationale of not processing cyto and syncytio separately?
- While the western blot did not show a difference in any ETC complexes other than complex IV, why the author’s measured respirations mediated by complex I and complex II? An explanation of these finding must be included in the discussion.
- The discussion of this manuscript lacks focus and doesn’t discuss or reason some important findings of this study. For instance, rather than discussing miR-210 (lines 311-320) and mt DNA (lines 342-352), it would be better to stay focused on the results if this study regarding the functioning of different complexes in ETC.
- As mentioned in line 332-333, low respiration rates are often associated with high ROS production (please also provide reference here), but the findings in this study are opposite. Authors have noticed low respiration as well as low ROS. How do you explain these results?
- The conclusion needs to be rewritten and drawn based on the results of this study. On the contrary, the most of conclusion is a sum up of conclusions of some previous studies (lines 364-377).
There are also some minor corrections:
Line 3 – define mt here.
Line 140-141 – further details of OXPHOS antibody cocktail should be provided including vendor, cat number and the concentration used.
Line 179 – abbreviate MAP here.
Line 225 – PE > 34
Line 318 – miR-210
Author Response
Reviewer 1:
- The introduction needs to be improved. As the study investigates the functionality of different complexes in ETC, a brief inclusion of functioning of ETC and unique roles of different complexes will be important for the readers to better understand the results and discussion.
Thank you for your comment. We have revised the introduction as you suggested and have added a section of ETC functioning and the roles of different complexes. On (lines 49-63).
- In the original protocol of isolation of trophoblast mitochondria, described by Fisher et al, they spun the supernatant at 4000 g for cytotrophoblast and 12000 g for syncytiotrophoblast isolation, respectively. However, the authors in this study used only 12000 g indicating that the pellet to isolate mt had both cyto and syncytiotrophoblast. What was the rationale of not processing cyto and syncytio separately?
Thank you for your comment. Our method was optimized from isolation protocol described by Bustamante et al (PMID: 25234730). Although the protocol describes two fractions, heavy mitochondria (possibly from cytotrophoblasts) and light mitochondria (possibly from syncytiotrophoblasts), our goal in the present study was to pellet mitochondria from whole placental tissue. We agree that processing cyto and syncytio fractions for bioenergetic profile would have captured differences in bioenergetics between these two trophoblast cell types, however, it was not our objective.
- While the western blot did not show a difference in any ETC complexes other than complex IV, why the author’s measured respirations mediated by complex I and complex II? An explanation of these finding must be included in the discussion.
Thank you for your comment. Respiration in intact isolated mitochondria is generally measured by the exogenous addition of substrates for complex I or complex II. Glutamate & malate provide NADH, and Succinate provide FADH2 which in turn provide a pair of electrons to complex I and II respectively (as the NADH or FADH2 reserves are very low in the isolated mitochondria). The impetus to measure complex I or II mediated respiration was not just measure complex I or II activities rather to assess respiration rate (which is dependent of all the complexes in ETC and complex V). Although, complex I or II activities were unchanged in our study, the reduced respiration rates indicate a compromised oxidative phosphorylation (possibly due to reduced complex IV activity). This has been revised in the discussion as suggested (lines 359-364).
- The discussion of this manuscript lacks focus and doesn’t discuss or reason some important findings of this study. For instance, rather than discussing miR-210 (lines 311-320) and mt DNA (lines 342-352), it would be better to stay focused on the results if this study regarding the functioning of different complexes in ETC.
Thank you for your comment. The Discussion is now revised as suggested and includes a clear discussion of the findings of this study.
- As mentioned in line 332-333, low respiration rates are often associated with high ROS production (please also provide reference here), but the findings in this study are opposite. Authors have noticed low respiration as well as low ROS. How do you explain these results?
Thank you for your comment. Low respirations are associated with high ROS production. For instance, any damage to electron transfer proteins downstream of complex I favor increased NADH/NAD+ ratio leading to back up of electrons at FMN site. These unused electrons react with available molecular oxygen within the matrix leading to ROS generation (PMID: 19061483). As explained in the discussion (lines 388-406), ROS measured in the study are hydrogen peroxide (H2O2). While H2O2 is one of the major ROS, it does not account for all reactive oxygen species. In fact, changes in metabolic pathways that regulate ROS can affect the levels of individual ROS. Hence, we cannot definitively state that PE placenta produce low ROS in comparison to NP placenta without assessing other forms of ROS.
- The conclusion needs to be rewritten and drawn based on the results of this study. On the contrary, the most of conclusion is a sum up of conclusions of some previous studies (lines 364-377).
Thank you for your comment. We have rewritten the Conclusions and ensured that they were drawn based on the results of our study.
- There are also some minor corrections:
Line 3 – define mt here.
Line 140-141 – further details of OXPHOS antibody cocktail should be provided including vendor, cat number and the concentration used.
Line 179 – abbreviate MAP here.
Line 225 – PE > 34
Line 318 – miR-210
Thank you for your comment. We have made the corrections to the lines above.

Reviewer 2 Report
Vaka et al. present the article “Characterization of Mitochondrial Bioenergetics in Preeclampsia” and overall this is an extremely interesting study. In my best knowledge study presents, for the first time, bioenergetics of isolated placental mitochondria in preeclamptic pathology. The study is novel and interesting but could be more structured and less chaotic.
Although the role of mt dysfunction from isolated placental mitochondria has been compared in preeclamptic women and normal controls in many previous studies, variations among preterm PE or term PE have not been compared. Surprisingly, the mechanisms contributing to mt ROS during PE are unclear. Although the authors did a great lab job, in my opinion, they have recruited a small number of women with preeclampsia and did not focus enough to compare both groups of preterm PE or term PE. as they intended to do so. Differences in PE<34 vs PE>34 need to be more clear so I suggest changing the structure of the paper.
Data from previous studies support the hypothesis that mt bioenergetics is impaired among preeclamptic patients which may be worse at preterm deliveries, so I wish to see the results and discussion on this field.
There are also some major concerns that need clarity to improve the quality of the study:
In Vaka’s study, the authors show that frozen placental mitochondrial membranes from PE patients exhibit significant reductions in complex IV activity and expression. However, they have not found any significant changes with other complexes of ETC, and this stays in contrast to other papers. If this happened because of technical problems (challenges) with the number and duration of freeze-thaw cycles, as they admit in the discussion, it makes me doubt the credibility of the other results! Could you comment on that?
I also support the thesis of the authors, that further studies exploring the factors involved in altering respiration and examining the relative contribution of major placental cell types to overall placental mt function could validate their findings and lead to new interventions that could be applied early in pregnancy and may yield positive outcomes in PE pathogenesis. But this is not the subject of this paper and is irrelevant for the paper. Don't you agree?
Vaka et al.'s findings show significant differences in mtROS generation between NP and PE patients. Low respiration rates are often associated with increased mtROS production. The existing literature on increased oxidative stress in PE placenta is very convincing, however, surprisingly this data indicates that PE<34 weeks placental mitochondria is significantly lower in comparison to PE>34 weeks placental. This needs more explanation.
The most important message from this study is, that therapies aimed to lower mtROS may not be of benefit to patients.
Author Response
Reviewer 2:
- Although the role of mt dysfunction from isolated placental mitochondria has been compared in preeclamptic women and normal controls in many previous studies, variations among preterm PE or term PE have not been compared. Surprisingly, the mechanisms contributing to mt ROS during PE are unclear. Although the authors did a great lab job, in my opinion, they have recruited a small number of women with preeclampsia and did not focus enough to compare both groups of preterm PE or term PE. as they intended to do so. Differences in PE<34 vs PE>34 need to be more clear so I suggest changing the structure of the paper.
Thank you for your comment. We have changed the structure of the paper to clearly provide details of our data showing the differences in preterm PE (PE<34) compared to PE>34 as well as the mechanisms contributing to mt ROS in the discussion.
- Data from previous studies support the hypothesis that mt bioenergetics is impaired among preeclamptic patients which may be worse at preterm deliveries, so I wish to see the results and discussion on this field.
Thank you for your comment. This section has been revised and the comment has been addressed in the discussion (Lines 536-946).
There are also some major concerns that need clarity to improve the quality of the study:
- In Vaka’s study, the authors show that frozen placental mitochondrial membranes from PE patients exhibit significant reductions in complex IV activity and expression. However, they have not found any significant changes with other complexes of ETC, and this stays in contrast to other papers. If this happened because of technical problems (challenges) with the number and duration of freeze-thaw cycles, as they admit in the discussion, it makes me doubt the credibility of the other results! Could you comment on that?
Thank you for your comment. It is possible that some of the differences in complex activity or expression across different research groups may be the result of variations in preparation and handling of mitochondria (i.e., number and duration of freeze thaw cycles). However, our prior optimization assays focused on validating mitochondrial membrane intactness, integrity, cytosolic contamination, and complex assays with appropriate controls (using chemical inhibitors of complexes) emphasize the validity of our techniques.
- I also support the thesis of the authors, that further studies exploring the factors involved in altering respiration and examining the relative contribution of major placental cell types to overall placental mt function could validate their findings and lead to new interventions that could be applied early in pregnancy and may yield positive outcomes in PE pathogenesis. But this is not the subject of this paper and is irrelevant for the paper. Don't you agree?
Thank you for your comment. This statement was not the subject of the paper, and was removed.
- Vaka et al.'s findings show significant differences in mtROS generation between NP and PE patients. Low respiration rates are often associated with increased mtROS production. The existing literature on increased oxidative stress in PE placenta is very convincing, however, surprisingly this data indicates that PE<34 weeks placental mitochondria is significantly lower in comparison to PE>34 weeks placental. This needs more explanation.
Thank you for your comment. This section has been revised and the comment has been addressed in the discussion. As explained in the discussion (Lines 928-946) research has showed that preterm PE placental mitochondria exhibit low H2O2 production in comparison to control pregnancies, and further showed that SOD expression in PE placental mitochondria was significantly low.